clinical trials; interventions; intervention response; longitudinal; precision medicine

**Author for correspondence:**
N. J. Schork,
Email: nschork@tgen.org

# Exploring human biology with N-of-1 clinical trials

N. J. Schork[1,2] ⬤, B. Beaulieu-Jones[2,3], W. S. Liang[2], S. Smalley[2,4] and L. H. Goetz[1,2] ⬤

[1]Department of Quantitative Medicine, The Translational Genomics Research Institute (TGen), Phoenix, AZ, USA; [2]Net. bio Inc., Los Angeles, CA, USA; [3]University of Chicago, Chicago, IL, USA and [4]The University of California Los Angeles, Los Angeles, CA, USA

## Abstract

Studies on humans that exploit contemporary data-intensive, high-throughput 'omic' assay technologies, such as genomics, transcriptomics, proteomics and metabolomics, have unequivocally revealed that humans differ greatly at the molecular level. These differences, which are compounded by each individual's distinct behavioral and environmental exposures, impact individual responses to health interventions such as diet and drugs. Questions about the best way to tailor health interventions to individuals based on their nuanced genomic, physiologic, behavioral, etc. profiles have motivated the current emphasis on 'precision' medicine. This review's purpose is to describe how the design and execution of N-of-1 (or personalized) multivariate clinical trials can advance the field. Such trials focus on individual responses to health interventions from a whole-person perspective, leverage emerging health monitoring technologies, and can be used to address the most relevant questions in the precision medicine era. This includes how to validate biomarkers that may indicate appropriate activity of an intervention as well as how to identify likely beneficial interventions for an individual. We also argue that multivariate N-of-1 and aggregated N-of-1 trials are ideal vehicles for advancing biomedical and translational science in the precision medicine era since the insights gained from them can not only shed light on how to treat or prevent diseases generally, but also provide insight into how to provide real-time care to the very individuals who are seeking attention for their health concerns in the first place.

## Impact statement

Individuals do not respond to health interventions in the same way. This creates a need for identifying what it is (e.g., a behavior, a gene, a biomarker, or their combinations) that may indicate which interventions should be provided to different individuals. In fact, a great deal of modern biomedical science has focused on the identification of the mechanisms that contribute to disease, and relevant research has revealed that most disease processes are indeed multifactorial and can differ substantially between individuals. However, only now are studies being pursued in earnest that seek to identify links between measurable factors and likely response to health interventions. In this light, studies designed to identify unequivocal individual responders and non-responders to health interventions are needed. Current approaches, specifically those involving large cohort-based clinical trials with single endpoints and a focus on average effects of an intervention, are not necessarily designed for this. Rather, emerging N-of-1 trial designs that focus on individual responses to an intervention by collecting enough data on a participant to statistically determine and quantify their responses are better suited for this. We provide the basic motivation and techniques used in N-of-1 studies, contrasting them with standard population-based clinical trials, and focus on directions in which the research community is going that could accelerate the use of strategies for providing health interventions to the individuals most likely to benefit from them. One key area where clinical studies of health interventions have fallen short is in limiting their focus on one health outcome or measure. It is underappreciated that what individuals put in their bodies may impact them in a wide variety of ways – both good and bad – and N-of-1 studies have the potential to help overcome this and thereby push the understanding of human biology in unprecedented ways.

## Introduction

The rapid development of high-throughput, cost-efficient and data-intensive assays for use in molecular biology and the biomedical sciences (e.g., DNA sequencing, proteomics, metabolomics, etc.) is revolutionizing the manner in which studies are pursued by seeking a deeper

understanding of the pathological processes underlying diseases of all sorts. The application of such technologies to, for example, explorations of the differences between diseased and non-diseased human tissue specimens or genome-wide association studies (GWAS) interrogating DNA collected on tens of thousands of individuals with and without a particular condition, has led to many very useful insights into how to combat diseases (Karczewski and Snyder, 2018). However, such investigations have also exposed one very complicated set of issues: most pathological processes underlying diseases are heterogeneous and nuanced, to the point where mechanisms contributing to disease in one individual may be different from those in another individual. Given this, it has also been shown that available treatments or preventive interventions for different diseases tend not to work in everyone with the same general diagnosis. These two facts have led to concerted efforts to promote 'precision' or 'personalized' medicine and nutrition whereby health interventions are tailored to the unique genomic, physiologic, clinical, behavioral and exposure profiles of individuals who could benefit from them (Ginsburg and Willard, 2016; Karczewski and Snyder, 2018; Zeggini et al., 2019).

The two largest impediments to enabling and deploying precision medicine at scale are (1) simply not having a more complete understanding of human in vivo biology and (2) not having insight into whether the differences exhibited by individuals at the molecular level – that have largely been identified from in vitro or ex vivo studies of human tissues – are truly clinically meaningful. Comprehensive longitudinal evaluations of humans using state-of-the-field assays have been pursued, but they have focused on identifying patterns among individuals in their natural environments without any controlled perturbation or design to relevant data collections (Chen et al., 2012; Li-Pook-Than and Snyder, 2013; Price et al., 2017; Earls et al., 2019; Schussler-Fiorenza Rose et al., 2019; Levy et al., 2020; Sailani et al., 2020; Zimmer et al., 2021; Metwally et al., 2022). Such studies are essential to explore human intra- and interindividual variation but leave open the question of how different factors might contribute to different responses to health interventions (Atkinson and Batterham, 2015; Atkinson et al., 2019; McInnes et al., 2021). We note that there are examples of specific therapeutic modalities whose development is consistent with and motivated by a precision medicine orientation in the discussion section.

The purpose of this review is to provide an argument that clinical trials can be pursued that will allow researchers to probe human physiology in ethically-sound ways with unprecedented sophistication. Relevant trials should be rooted in N-of-1 and aggregated N-of-1 designs (Schork, 2015; Nikles et al., 2021) and focus on exploring multiple phenotypes simultaneously and identifying causal relationships between phenotypes by leveraging emerging, largely non-invasive, health monitoring devices and assays (Izmailova et al., 2018; Bentley et al., 2019; Tehrani et al., 2022). We do not provide an exhaustive review of N-of-1 trials, as there are many excellent resources and introductions to the basic motivation and methodologies (Lillie et al., 2011; Nikles et al., 2021; Davidson et al., 2022), including comprehensive reviews of the applications of N-of-1 trials (Gabler et al., 2011; Li et al., 2016; Mirza et al., 2017) as well as practical guides as to how to conduct N-of-1 trials (Guyatt et al., 1988; Kravitz et al., 2014; Nikles et al., 2021; Duan et al., 2022). In fact, N-of-1 trials are now receiving attention as strategies for improving health care generally (Keller et al., 1988; Senn, 1998; Derby et al., 2021; McDonald and Nikles, 2021; Selker et al., 2022). Rather, we focus on N-of-1 trials that can

address issues plaguing precision medicine and can provide a better understanding of human biology for at least four reasons: (1) They can provide unprecedented insights into human biology, including intra-individual causal claims about interventions and health measures. (2) They provide very comprehensive ways of vetting interventions to see if they work and for whom they work. (3) Their results provide insight into an individual's health that may benefit them almost immediately, as opposed to much later after all relevant data have been collected and analyzed as part of a larger study. (4) Their results can be aggregated to explore patterns among individuals who exhibit robust responses to interventions. The organization of the review is as follows. We first provide greater insight into why legacy population-wide effect-focused randomized clinical trials (RCTs) are inadequate to address fundamental questions about human biology. We then consider different aspects of, and settings for, the proposed multivariate N-of-1 clinical trials, including the need for better markers of drug activity and availability. We end with a brief discussion of a few emerging therapeutic areas that could benefit from the proposed trials as well as suggestions for future research.

## Human biology and legacy clinical trials

Strategies to understand how systems function as a whole, and which components may be dependent on other components, typically involve inducing perturbations to those systems and then determining how the systems respond (e.g., in cellular or mouse physiology studies). Studies seeking to perturb living humans systematically in this way are at worst unethical and at best logistically complicated. However, humans voluntarily subject themselves to perturbations of all sorts via pharmacologic interventions, dietary manipulations, environmental exposures, etc. In fact, clinical trials are routinely pursued to explore responses to such perturbations. Unfortunately, most clinical trials tend to focus on a singular indication (i.e., health or response measure) and the average response to the intervention in the population at large and therefore do not address broader questions about human physiology. We do not provide an in-depth review of clinical trials here (see, e.g., Friedman et al., 2015), but rather highlight a few of their key aspects so they can be contrasted with the N-of-1 studies. Typically, health interventions are evaluated in stages to ensure their safety and efficacy, from small ($n = 5$–$20$) phase I safety trials, to moderately sized ($n = 25$–$200$) phase II efficacy trials, to large ($n = 250$–$10,000$) phase III comparative and phase IV post-marketing surveillance studies. Some phase II and virtually all phase III and IV trials are pursued as RCTs where individuals are randomized to receive or not receive the intervention in question to avoid confounding. The health measures collected on these individuals are then compared to determine what effect the intervention may have on the typical person in the population at large.

There are at least six issues in the conduct of phase I–phase IV clinical trials (Deaton and Cartwright, 2018; Schork, 2018) that motivate complementary N-of-1 trials: (1) Most standard clinical trials have inclusion and exclusion criteria to make sure the trial has been carried out in individuals likely to benefit, as well as for ensuring safety and avoiding confounding effects, which can complicate their generalizability. (2) Most, if not all, trials focus on the effect of an intervention on a single well-defined endpoint (e.g., such as blood pressure, pain, or rheumatoid arthritis symptoms). (3) Most failures of interventions in clinical trials testing occur in the phase II stage of testing; that is, despite being shown to have

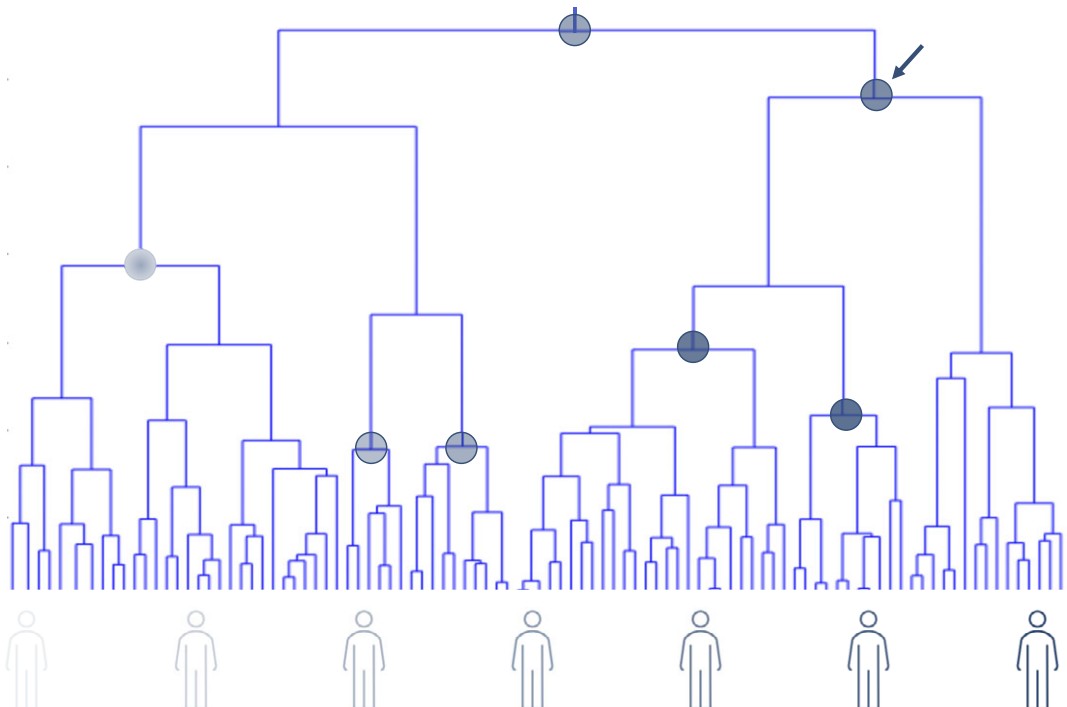

**Figure 1.** A tree or dendrogram reflecting how similar a number of individuals are with respect to phenotypes of relevance to drug response: the closer the bottommost branches of Figure 1 are – which represent individuals – the more similar the phenotypic profiles of those individuals are. The darkness of the shaded human figures at the bottom of the figure at different positions in the tree reflects the degree to which individuals at those positions in the tree possess a certain characteristic or profile. The circles represent interventions that can benefit different groups of individuals, such that the different locations where the shaded circles are situated represent convergence points for all individuals connected beneath that point who can benefit from the specific intervention. Thus, the topmost circle indicates that all individuals may benefit from that intervention (since all the individual tree branches converge back to that point), whereas the leftmost circle is likely to benefit the first ~25–30% of individuals. The two circles second and third from the left indicate interventions that may benefit a small number of individuals (e.g., only ~10% of individuals). The circle to which the arrow is pointing indicates an intervention that may benefit a large number of individuals but for whom other interventions (reflected by the 5[th] and 6[th] circles from the right) may benefit smaller subsets of individuals. Identifying points on trees like this that are consistent with who benefits from an intervention based on understanding of the factors responsible for mediating response is the motivation behind precision medicine and nutrition.

potential in 'pre-clinical' cellular and non-human experiments and to be safe in phase I trials, many interventions are shown not to modulate or affect the phenotype they were designed to impact, calling into question the pre-clinical, basic-science driven evidence suggesting that they may have benefit in humans *in vivo* (of course there are other reasons why an intervention may fail in a Phase II trial, for example, due to biased sampling, focusing on the wrong endpoint, measurement error, etc.). (4) Most late phase clinical trials, despite having inclusion and exclusion criteria, are expensive as they are conducted on very large numbers of people to ensure the trial results are generalizable and to overcome often hypothesized weak average effect sizes. (5) The results of clinical trials may identify interventions with the potential to benefit individuals, but unless it is known *a priori* how to identify individuals most likely to benefit from each intervention, it will be unclear how to optimally provide the interventions (see Figure 1). (6) Standard population-based RCTs can take a very long time to pursue and analyze, whereas more focused participant or patient-oriented alternative trial designs can be aggregated sequentially to enable population-based inferences (Schork, 2022).

## Basic N-of-1 trial designs

### Basic designs

As emphasized, the ultimate goal of N-of-1 trials is to determine, in an appropriately powered way, if an intervention is actually benefitting a target individual by leveraging data collections and analytical methods focused on that target individual's response. An element common to all N-of-1 clinical trial designs is an intervention 'crossover' component in which measurements on a health-related phenotype (e.g., blood pressure, mood, weight, symptoms, etc.) are made while the target individual is receiving, and not receiving, an intervention. This contrast between measures while on and off the intervention can then be exploited to quantify and characterize the individual's response to the intervention but only if enough reliable measurements are made during each of the intervention periods and data analysis methods are used to control for confounding due to, for example, placebo or unmeasured covariate effects (Lillie et al., 2011; Kravitz et al., 2014; Wang and Schork, 2019; Kravitz and Duan, 2022). Note that many of the most widely used strategies for avoiding confounding in standard RCTs can be exploited in the design and execution of N-of-1 trials, such as randomizing the order in which the interventions are provided, blinding of the received interventions to the participants and/or researchers analyzing the data, washout periods to avoid carryover effects, etc. (Lillie et al., 2011; Duan et al., 2013; Kravitz et al., 2014; Duan et al., 2022; Kravitz and Duan, 2022).

Figure 2 depicts some basic N-of-1 designs. We note that there is growing, but not complete, consensus on the definition of an N-of-1 clinical trial – which many believe requires a randomized order of interventions with, for example, blinding – as opposed to a simple 'single case study' which may not include randomization or blinding. We argue that both N-of-1 clinical trials and some single case

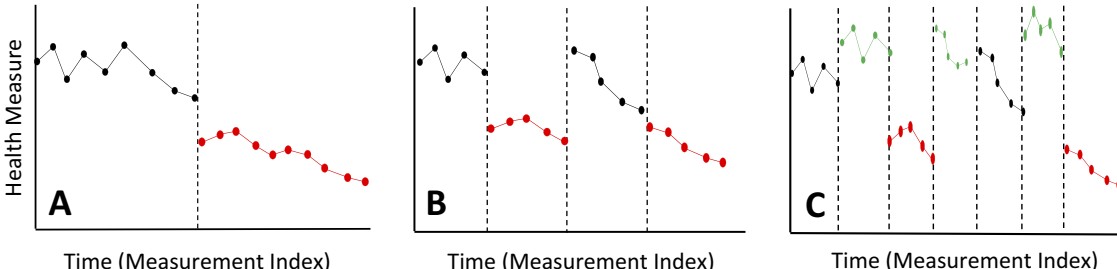

**Figure 2.** Different, very basic, types of N-of-1 clinical trial designs in which an intervention had a lowering effect on a health measure (like blood pressure). The black and red lines reflect hypothetical health measure trajectories (i.e., longitudinal data) while an individual is not receiving (black) or receiving (red) an intervention. The vertical dashed lines indicate when interventions were provided or changed. Panel A depicts the basic 'interrupted time' series design, Panel B the 'reversal' design and panel C a reversal design with washout periods (green lines).

studies are appropriate for advancing precision medicine (Davidson et al., 2022) and consider them both as N-of-1 clinical trials. Panel A depicts the simple and often used 'interrupted time series single case design' – or basic 'AB' design, where 'A' and 'B' correspond to interventions, one of which could be a placebo or simply no intervention (see, e.g., part V of the book by Huitema, 2011 for an excellent introduction). Panel B depicts the 'reversal' or 'ABAB' design in which the intervention periods in the interrupted time series design are repeated to ensure the initial set of observations do not reflect false positive or negative results. Panel C depicts the reversal design with washout periods (i.e., periods where no administration of an intervention, including a placebo, are provided) between each administration of an intervention to avoid confounding carryover effects (an 'AwBwAwB' design). Note that the number of intervention administration periods and the order of the interventions can vary depending on the sophistication of the design (e.g., 'ABwBA' or 'AwAwBwAwBwBwA').

### The power of N-of-1 trials

N-of-1 trials derive their power to make inferences about the effect of an intervention on an individual from the number of measurements made on the participant while on and off an intervention (Huitema, 2011). However, serial correlations between the measurements can complicate the analysis if not appropriately accounted for, as can aforementioned covariate effects, carryover effects, missing data, non-uniform time points between measurement collections and placebo effects (Rochon, 1990; Huitema, 2011; Lillie et al., 2011; Wang and Schork, 2019; Somer et al., 2022). Many offshoots of N-of-1 trials exist to improve their efficiency and comprehensiveness; for example, sequential designs can be used to minimize the number of measurements made while preserving appropriate false positive and false negative rates (Schork and Goetz, 2017; Schork, 2022). In addition, there is no reason that N-of-1 trial methodology cannot be used in other settings, for example, assessing intervention effects in cell lines, tissue samples, mice, etc. In fact, such studies often make use of samples from a single individual or strain of mice and so, from a biological standpoint, they are, by their nature, assuming that insights from a single individual can shed light on very general biological questions. There are many recent examples of N-of-1 studies, which we will not review exhaustively here (Gabler et al., 2011; Kronish et al., 2018; Nikles et al., 2022; Samuel et al., 2022), but rather simply emphasize that they are growing in number and sophistication (Kim et al., 2019; Lamb et al., 2022; Phyland et al., 2022).

### Beyond the basics

There are three important aspects of N-of-1 trials that are receiving the attention which are motivating newer approaches. First, the data and results associated with individual N-of-1 trials can be aggregated and analyzed to explore trends among the participants and their responses (Zucker et al., 2010; Araujo et al., 2016; Punja et al., 2016; Schork and Goetz, 2017; Barbosa Mendes et al., 2022). Second, with sufficient data collected over time, one could characterize *causal* relationships among the intervention and other measures (Molenaar, 2019; Izem and McCarter, 2021; Yeboah et al., 2021) (note: an entire recent issue of the journal 'Evaluation and the Health Professions' was devoted to causal analysis in N-of-1 trials (Miocevic et al., 2022). Such analyses could provide unprecedented insight into human physiology. The third is that the execution of N-of-1 trials focusing on important physiologic endpoints can be greatly enhanced with emerging digital health-based monitoring devices (such as the Apple Watch and continuous glucose monitors), survey instruments made available through smartphone apps, and largely pain-free and convenient methods for obtaining blood, urine, stool and saliva samples (Enderle et al., 2016; Izmailova et al., 2018).

### Multivariate n-of-1 trials

N-of-1 clinical trials can be pursued to characterize the effect of an intervention on a specific phenotype (blood pressure) for a target individual and as such complement population-based RCTs, especially when it is unclear if an individual is likely to benefit from the intervention. However, many diseases are not associated with singular phenotypes and, in fact, most individuals who suffer from them do not only have one major symptom or problem (Ong et al., 2020). This is especially the case for older individuals with many comorbidities (Pearson-Stuttard et al., 2019; Onder et al., 2020; Skou et al., 2022). As a result, it makes sense to pursue appropriately powered N-of-1 trials that explore the impact of an intervention on more than one outcome (i.e., multivariate N-of-1 trials). Although multivariate trials have been proposed in the context of standard RCTs, there are few, if any, precedents in N-of-1 study contexts (Zhao et al., 2009). Few published precision medicine studies have measured more than one clinically relevant health measure despite the availability of newer health monitoring technologies (Viana et al., 2021). Although we will not go into the mathematical or statistical details here for how such trials can achieve sufficient power, it is arguable that if health is defined broadly (e.g., normal blood pressure, quality sleep, good blood biochemistry profile, etc.)

then a good health intervention should at a minimum not negatively affect any of them and at best positively affect them all. In this light, testing multiple measures for intervention effects simultaneously using an omnibus statistical test of the hypothesis that an intervention positively effects them all could lead to an increase in power (Huitema, 2011; Tabachnick and Fidell, 2012), but only if the number of measures is large (Leroy et al., 2022). Reaching appropriate numbers of observations could be achieved, for example, through the use of the aforementioned continuous wireless devices or microsampling techniques which involve collecting minute amounts of blood or urine for analyses to avoid a standard blood draw or logistically challenging biospecimen collections (Enderle et al., 2016; Bentley et al., 2019; Anderson et al., 2020).

There are many settings beyond multimorbidity issues that justify an evaluation of multiple health measures in N-of-1 clinical trials. For example, depression is known to impact virtually all aspects of a person's health due to the various behaviors adopted by depressed individuals (Triolo et al., 2020; Aprahamian et al., 2022). Testing the effect of an antidepressant on mood and depressive symptoms in addition to, perhaps, weight, blood pressure, sleep quality, etc. makes sense. Another example involves geroprotectors, or interventions meant to slow the aging rate and thereby influence susceptibility to, or processes associated with, many different age-related diseases (Mahmoudi et al., 2019; Kritchevsky and Justice, 2020; Triolo et al., 2020; Aprahamian et al., 2022; Moskalev et al., 2022). Thus, by definition, a geroprotector should affect multiple systems and hence could be tested for this. In fact, if only one or some subset of health measures among many different measures is in fact affected by a purported geroprotector, then the intervention is probably not a geroprotector (Schork et al., 2022).

In addition to testing for the effect of an intervention on multiple health measures, N-of-1 and aggregated N-of-1 studies can be pursued to exploit interventions as ways of perturbing or probing human physiology – the goal being to identify relationships among different health measures or processes. Thus, if enough measures are collected over the time an individual is both receiving and not receiving an intervention, then temporal relationships between the measures can reveal likely causal relationships among them based on, for example, time series analysis, Granger regression and other techniques (McCracken, 2016; Molenaar, 2019). Such analyses would again be significantly enhanced if the relevant health measures were collected continuously (Enderle et al., 2016; Bentley et al., 2019; Anderson et al., 2020). In addition, by assessing the effect of the intervention on health measures beyond a primary measure in relevant trials, potential intervention 'repurposing' opportunities could arise (Pushpakom et al., 2019; Krishnamurthy et al., 2022; Mucke, 2022). In this way, N-of-1 trials can be pursued as proof-of-concept studies for identifying multiple indications, or at least one on solid footing, for an intervention (Pushpakom et al., 2019; Mucke, 2022). In addition, by collecting multiple health measures on an individual N-of-1 trial participant, possibly continuously and in real time, insights into that participant's health and health trajectory can be obtained even if an intervention being tested is shown not to benefit the participant.

## Whole body, biomarker validation and therapeutic drug monitoring studies

There are some very specific areas where multivariate N-of-1 trials can be pursued that will enhance the assessment of individual

intervention response and enable deeper insight into human physiology, as emphasized throughout this review. We briefly describe four such areas below.

### General assessment of inter-individual variation in intervention response

As noted, given that N-of-1 trials focus on individuals' responses, they can be used to more precisely identify responders to particular interventions. In addition, if relevant studies collected sufficient data on more than one health measure then they can be used to identify potential side effects, alternative uses for the intervention and different mechanisms of action or physiological processes modulated by the intervention. In fact, it might make sense for all interventions to be evaluated for their whole-body effects in a small number of individuals as they are being developed. If done along the lines outlined in the review, such trials could shed enormous light on how substances put into the human body affect it systemically (see Figure 3).

### Biomarker and surrogate endpoint validation

There is great interest in identifying better biomarkers of an intervention's activity so that these biomarkers can be correlated with other health measures of interest (see, e.g., 'Therapeutic Drug Monitoring Studies' section below) (Hendrickson et al., 2020). In addition, there is also interest in identifying 'surrogate endpoints' for clinical trials that initially focus on expensive, lengthy and logistically challenging health outcome measures, and N-of-1 trials are excellent vehicles for validating biomarkers and surrogate endpoints (Burzykowski et al., 2005). As an example, consider the development and use of epigenetic clocks as surrogate endpoints in trials of geroprotectors (Schork et al., 2022). The belief is that if an intervention modulates or changes an epigenetic clock among participants in a trial in positive ways – thereby indicating that the intervention in question is slowing the aging rate of the individuals – then those individuals do not necessarily have to be tracked longitudinally until they develop (or do not develop) age-related diseases that the candidate geroprotector is hypothesized to prevent or treat (Mahmoudi et al., 2019; Kritchevsky and Justice, 2020; Schork et al., 2022). Thus, the epigenetic clocks would act as a surrogate endpoint for the processes that are associated with the disease endpoints of real interest, which are modulated by the intervention. Although epigenetic clocks have been shown to be correlated with disease endpoints, they have been done so via large epidemiological studies and not in focused clinical trials measuring appropriate health measures. Therefore, it is arguable that by measuring epigenetic clocks along with health measures that underlie many common chronic age-related diseases and conditions, such as blood pressure, cholesterol level, sleep quality, etc. in appropriately powered N-of-1 trials, one might not only show that the geroprotector influences these health measures in positive ways, but also that an epigenetic clock is correlated with them as well. This would in effect validate surrogacy of the epigenetic clock at the 'level of the individuals and the trial' (Burzykowski et al., 2005; Buyse et al., 2022).

### Therapeutic drug monitoring studies

Therapeutic drug monitoring (TDM) studies consider the measurement of a drug's concentration in an individual's bloodstream in

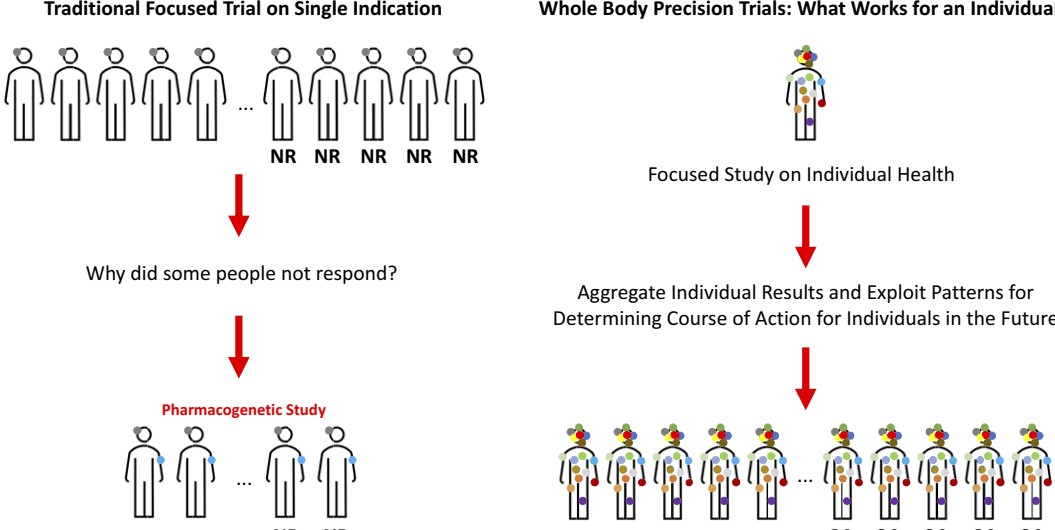

**Figure 3.** Contrasting clinical trial designs. The design depicted on the left is consistent with standard RCTs focusing on a singular health measure or indication (the gray colored dot on the left side of the head of the human figures indicating a single phenotype of interest; for example, depression symptoms). If individuals are found not to respond (NR = Non-Responders) then a future study seeking to identify biomarkers of response could be pursued, whereby a new biomarker phenotype is associated with the response/non-response phenotype (e.g., genomic profile). The design depicted on the right provides the motivation for complementary trials to traditional RCTs whereby the effect of an intervention is evaluated on an individual from a whole-body perspective. The results of this trial are aggregated with trials on other individuals and patterns that could identify responders and non-responders are explored that may also reveal intervention effects on different phenotypes and how those phenotypes interact.

order to correlate the levels of the drug with the phenotype that the drug is hypothesized to modulate (Dasgupta, 2012; Clarke and Dasgupta, 2019). Most drugs do not undergo such evaluation and testing, which is unfortunate since such studies could in theory better characterize mechanisms of action of the drug and its effects on different phenotypic endpoints. Of course, TDM studies are predicated on the assumption that there is a definable relationship between drug dose and plasma or blood drug concentration, and between concentrations and therapeutic effects. In addition, TDM studies require ways of measuring blood levels of a drug which may not be trivial. However, by more precisely measuring drug bioavailability and activity in N-of-1 trials, especially in trials for which participants are monitored for multiple health measures, one could explore temporal relationships between drug bioavailability and activity and not just, for example, pill count-based dosing and outcomes (Dasgupta, 2012; Clarke and Dasgupta, 2019; Irving and Gecse, 2022; Ordutowski et al., 2022).

### *Matching based on data aggregation*

As noted previously, if enough N-of-1 trials are pursued using the same interventions, and baseline health assessments with common measures have been collected on each participant, then the data and results can be aggregated and analyzed. The common baseline health examination profiles of the individuals could then be explored for patterns and correlations with intervention responses. This can enable matching a future target individual's baseline health profile with others' profiles who previously went through N-of-1 trials. If good matches (however defined) are found, then the interventions to which those individuals matching the target individual responded, would be reasonable first-choice interventions for the target individual (Wicks et al., 2011; Schork and Goetz, 2017; Schork et al., 2020; Davidson et al., 2022). Different strategies for identifying the matches could be pursued based on, for example, propensity scores and related techniques (Guo and Fraser, 2014; Liu and Meng, 2016).

### Conclusions and future directions

There are few health interventions whose effectiveness is ubiquitous. This can be attributed to the great genetic, physiologic, clinical, behavioral and exposure profile variation exhibited by individuals susceptible to or suffering from diseases (Schork, 2015). Identifying interventions that benefit individuals on the basis of their nuanced and possibly unique profiles is the goal of precision or personalized medicine. However, tailoring or matching interventions to individuals will require greater understanding of intra- and inter-individual variation and intervention response and, as argued throughout, can be enabled or enhanced through the use of whole-body N-of-1 clinical trials (Figures 1 and 3).

In this light, many emerging interventions, such as cytotoxic T-cell therapies (Kiyotani et al., 2021; Roesler and Anderson, 2022), brain anatomy-guided Transcranial Magnetic Stimulation (TMS) therapies (Siddiqi et al., 2021; Williams et al., 2021) and sequence-based antisense oligonucleotide therapies (Kim et al., 2019; Helm et al., 2022), are designed to only work on specific individuals given that the targets they exploit and constructs they use are based on the unique features underlying the pathologies of the individuals for whom they are designed. Testing the effectiveness of these interventions, given that no two individuals with the same condition will likely get exactly the same intervention, could make use of the proposed N-of-1 strategies. Of course, one could address very broad questions about the utility of such interventions using standard RCTs, such as whether individuals who receive the personalized interventions fare better than individuals who receive a more 'one-size-fits-all' intervention (Schork et al., 2020).

Ultimately, the current emphasis on precision medicine, the emergence of sophisticated health monitoring technologies, and the desire of individuals to optimize their health and not simply contribute to studies that may only benefit future generations, demand better approaches to biomedical and translational science. We recognize that there might be impediments to the implementation of multivariate N-of-1 trials of the type described. For

example, a greater patient burden for data collection, logistical complications in collecting different data types, and the costs of conducting and monitoring the individual participants may create barriers to the adoption and use of multivariate N-of-1 trials. However, efficient, cost-effective and participant-friendly N-of-1 clinical trials – to the degree that they can be pursued – are very likely to be an appropriate addition to biomedical and translational studies in the future given that they have at least 4 very overt advantages, including: (1) the ability to shed light on fundamental questions about human biology; (2) determine which interventions work and on whom; (3) benefit the participants in the trials directly and almost immediately by collecting vast amounts of health data on them possibly continuously and with real-time interpretive ability; and (4) pave the way for their aggregation and analysis to identify patterns that may inform their use and execution in the future.

**Open peer review.** To view the open peer review materials for this article, please visit http://doi.org/10.1017/pcm.2022.15.

**Acknowledgements.** The authors would like to thank Drs. Mark Adler and Stephanie Venn Watson for commenting on earlier versions of this manuscript.

**Author contributions.** N.J.S. conceived of the orientation and format for the review, and N.J.S., B.B.-J., W.S.L., S.S. and L.H.G. pursued relevant literature reviews. N.J.S. wrote the initial draft, L.H.G. edited the initial draft and B.B.-J., W.S.L. and S.S. edited the subsequent drafts.

**Financial support.** N.J.S. is supported in part by the following grant support from the National Institutes of Health: 1 U19 AG056169-01A1; U2C CA252973; UH3 AG064706; UH2 AG06470602S1 and U19 AG023122.

**Competing interest.** N.J.S., S.S. and L.H.G. are founders of net. Bio, a company focusing on pursuing novel clinical protocols to ensure that individuals benefit from health interventions of all sorts. B.B.-J. and W.S.L. are paid consultants for net. Bio.

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
