## [Reviewer Report]

*Comments to Author*: The authors of this manuscript deliver an excellent commentary on what N-of-1 trials have to offer precision medicine. They share a vision for the role of multivariate N-of-1 trials and provide a compelling argument for why such trials can provide a valuable contribution to the evidence base in many clinical areas. The authors make the case for multivariate N-of-1 trials addressing fundamental unanswered questions about human biology and being used as a tool to facilitate drug repurposing, which has received considerable attention in the scientific literature of late (e.g. Pushpakom et al., 2019; Krishnamurthy et al., 2022)

There are two issues I think the authors could consider addressing in a revision of the manuscript:

(1) Some single-case designs mentioned briefly in this paper (AB, ABAB), and visually represented in Figure 1, are not typically considered an “N-of-1 trial” design. I think it is generally accepted that N-of-1 trials are those that involve multiple crossovers that are randomly determined (+/- blinding) and therefore mention of the AB and ABAB design may not be needed. Alternatively, these designs could be described under the broad umbrella term “single-case designs”.

(2) I think it would be appropriate to outline some challenges for multivariate N-of-1 trials; they will require frequent measurement of all outcomes of interest to achieve the statistical power needed for the analysis but some outcomes may not be amendable to such frequent measurement (yet). A couple of sentences to briefly acknowledge these challenges may be useful. In addition, it may not be clear to the reader what the authors mean by doing N-of-1 trials “properly”. Perhaps some extra detail here to clarify would also be useful.

Some other minor points the authors could consider are listed below.

• Although in N-of-1 trials patients may not have to wait for as long as they do in RCTs to receive any results shared with them, it is not clear from the paper how using N-of-1 trials enable them to receive “real time” care (mentioned in the abstract). There may be statistical techniques/packages and technology that can fast-track or automate the analysis of N-of-1 trial data, but it might be good to cover this briefly in the manuscript to endorse the possibility of delivering real-time care.

• The authors state: “Most standard clinical trials have inclusion and exclusion criteria to make sure the trial has been carried out in individuals likely to benefit” (page 9) – other reasons could include safety reasons, to avoid confounds and to ensure the individuals are similar so the results can apply to this group of individuals.

• The authors state: “many interventions are shown not to modulate or affect the phenotype they were designed to impact, calling into question the ‘pre-clinical,’ basic-science driven evidence suggesting that they may have benefit in humans in vivo” (page 9) – this could happen for other reasons also.

• A fifth point to extend the section on page 9-10 could be the lag time to obtain results.

• It may not be clear to the reader what is meant by micro-sampling techniques (page 14). Is this questionnaire sampling methods like ecological momentary assessments or something else? Perhaps the authors could add one or two examples in parenthesis.

• The second paragraph in the conclusion (page 19) might fit better in the main manuscript as an additional future direction/opportunity. ASO is an exciting area that perhaps deserves more space?

• In the last paragraph in the conclusion (page 20) “N-of-1 trials have a 4 fold advantage” - I wondered what you were comparing them to in this statement, as there are possibly many more than just four advantages (depending on what the comparator is).

• If appropriate, the authors may wish to consider mentioning the International Collaborative Network for N-of-1 Trials and Single-Case Designs (www.nof1sced.org) as a resource for those interested in this design (Nikles, J., Onghena, P., Vlaeyen, J. W., Wicksell, R. K., Simons, L. E., McGree, J. M., & McDonald, S. (2021). Establishment of an International Collaborative Network for N-of-1 Trials and Single-Case Designs. Contemporary clinical trials communications, 23, 100826).

---

## [Reviewer Report]

*Comments to Author*: In this review, the authors wrote about exploring human biology with N-of-1 clinical trials. This is an interesting topic. The comments from this reviewer are as follows:

1. In Abstract, the authors should clearly state the purpose of the review, so that the readers could understand the content and structure of this review better.

2. The main text of the review included six parts: Introduction, Human biology and legacy clinical trials, Basic N-of-1 trial designs, Multivariate N-of-1 trials, Whole body, biomarker validation, and therapeutic drug monitoring studies, Conclusions and future directions. The authors should clearly state the purpose and the structure of the review in the first part.

3. In the paragraph of ‘Therapeutic drug monitoring studies’, the authors wrote ‘However, by more precisely measuring drug bioavailability and activity in N-of-1 trials, especially in trials for which participants are monitored for multiple health measures, one could explore temporal relationships between drug bioavailability and activity and not just, e.g., pill count-based dosing and outcomes.’ Please give reference(s) for this sentence.

4. For Figure 1 and Figure 2, although Figure legends were given, it is recommended to add symbol notes in the figures as well.